# SKF-96365 Expels Tyrosine Kinase Inhibitor-Treated CML Stem and Progenitor Cells from the HS27A Stromal Cell Niche in a RhoA-Dependent Mechanism

**DOI:** 10.3390/cancers16162791

**Published:** 2024-08-08

**Authors:** Audrey Dubourg, Thomas Harnois, Laetitia Cousin, Bruno Constantin, Nicolas Bourmeyster

**Affiliations:** 1UMR 6041 CNRS/Université de Poitiers, “Channels and Connexins in Cancer and Cell Stemness”, Pôle Biologie Santé, 1, rue Georges Bonnet, 86021 Poitiers CEDEX, France; audrey.dubourg@univ-poitiers.fr (A.D.); thomas.harnois@univ-poitiers.fr (T.H.); laetitia.cousin@univ-poitiers.fr (L.C.); bruno.constantin@univ-poitiers.fr (B.C.); 2CHU de Poitiers, Pôle BIOSPHARM, Secteur Biochimie, 86022 Poitiers CEDEX, France

**Keywords:** Chronic Myeloid Leukemia, leukemia stem cells, RhoGTPases, tyrosine kinase inhibitors, SKF-96365, leukemia niche

## Abstract

**Simple Summary:**

Tyrosine kinase inhibitors (TKIs), particularly imatinib, have afforded Chronic Myeloid Leukemia (CML) patients long-term remission. This revolution in leukemia treatment is unfortunately accompanied by the fact that treatment interruption, in most cases, leads to the re-emergence of BCR-ABL-leukemia cells, caused by leukemia stem cells, that are insensitive to TKI treatment. Moreover, TKIs paradoxically induce a quiescent state in leukemia stem and progenitor cells. Current CML studies focus on these quiescent leukemia stem cells in order to reactivate them and restore TKI sensitivity. We show here that under TKI treatment, leukemia stem cells retrieve motility in a reconstituted niche when incubated with SKF96365, a calcium channel inhibitor, and that this effect is dependent on RhoA activation by BCR-ABL independently of its tyrosine kinase activity. Altogether, these results suggest that SKF-96365 or its derived molecules could be interesting compounds for targeting quiescent CML stem and progenitor cells under TKI treatment.

**Abstract:**

Background: A major issue in Chronic Myeloid Leukemia (CML) is the persistence of quiescent leukemia stem cells (LSCs) in the hematopoietic niche under tyrosine kinase inhibitor (TKI) treatment. Results: Here, using CFSE sorting, we show that low-proliferating CD34+ cells from CML patients in 3D co-culture hide under HS27A stromal cells during TKI treatment—a behavior less observed in untreated cells. Under the same conditions, Ba/F3p210 cells lose their spontaneous motility. In CML CD34+ and Ba/F3p210 cells, while Rac1 is completely inhibited by TKI, RhoA remains activated but is unable to signal to ROCK. Co-incubation of Ba/F3p210 cells with TKI, SKF-96365 (a calcium channel inhibitor), and EGF restores myosin II activation and amoeboid motility to levels comparable to untreated cells, sustaining the activation of ROCK. In CFSE+ CD34+ cells containing quiescent leukemic stem cells, co-incubation of TKI with SKF-96365 induced the expulsion of these cells from the HS27A niche. Conclusions: This study underscores the role of RhoA in LSC behavior under TKI treatment and suggests that SKF-96365 could remobilize quiescent CML LSCs through reactivation of the RhoA/ROCK pathway.

## 1. Introduction

Chronic Myeloid Leukemia (CML) is characterized by the presence of a translocation between chromosomes 9 and 22 in the hematopoietic stem cell, leading to the expression of the chimeric protein BCR-ABL. The constitutive tyrosine kinase activity of BCR-ABL is responsible for the expansion of myeloid progenitors, and for the last 20 years, tyrosine kinase inhibitors (TKIs) have provided a very efficient treatment for CML [1]. One of the major problems encountered in CML treatment is that some leukemia stem cells (LSCs) and immature progenitors are insensitive to TKIs and persist into the hematopoietic niche, representing a pool of malignant cells responsible for disease recurrence in cases of TKI withdrawal [2,3]. Furthermore, recent publications show that TKIs are responsible for driving these LSCs into quiescence [4,5]. Many studies have tried to find alternative ways to eradicate these CML LSCs [6,7,8,9,10], and although some studies have shown interest in RhoGTPases for CML treatment [11], none have focused on the potential of RhoGTPases in targeting stem cells [12].

For many years, we have studied the role of RhoGTPases in BCR-ABL signaling, particularly from the perspective of migration. We demonstrated that the DH/PH domain of p210^BCR-ABL^ specifically activates RhoA, while Rac1 and Cdc42 are indirectly activated by Vav isoforms, which are activated by BCR-ABL-operated tyrosine phosphorylation. Rac1 triggers the motility of BCR-ABL-expressing cells, while RhoA specifically activates the ROCK1 isoform, which phosphorylates myosin light chain, allowing the amoeboid mode of migration [13]. Since the DH/PH domain of BCR-ABL is independent of its tyrosine kinase domain, one could hypothesize that RhoA activity could be conserved during TKI treatment. In hematopoietic stem and progenitor cells, the RhoA/ROCK pathway was shown to be essential for expansion and survival [14,15]. Until now, in the case of CML LSCs, no study has examined the role of the RhoA/ROCK pathway. The enhancement of motility, leading to the exiting of quiescent LSCs from the protection of the hematopoietic niche, could represent an interesting approach to targeting these resilient cells. We aimed to determine the motility behavior of quiescent LSCs from CML patients under TKI treatment in comparison to Ba/F3p210 cells. We investigated the activation state of RhoGTPases under TKI treatment in the Ba/F3 cell line and in CD34+ CML cells. Finally, we explored the effects on motility of a calcium channel inhibitor, SKF-96365, which we have previously shown to enhance the motility of BCR-ABL-expressing 32D cells [16]. This inhibitor was then used to try to remobilize imatinib-induced quiescent LSCs from CML patients.

## 2. Materials and Methods


**Cell lines**


The Ba/F3- and BCR-ABL-expressing cell lines Ba/F3p210 and Ba/F3p210S509A were cultured as described previously [17]. The stable recombinant Ba/F3p210 and Ba/F3p210S509A cell lines were generated as previously described [17]. The HS27A cell line was cultured similarly to the Ba/F3 cell lines.


**Isolation and culture of CD34+ CML progenitor cells**


Collaboration with Poitiers Centre Hospitalier Universitaire CRB allowed access to blood and bone marrow samples of CML patients at diagnosis, obtained after regular consent was collected by the CRB. Mononuclear cells were isolated from BM samples with Ficoll density gradient centrifugation (25-072-CV, Corning, NY, USA). The CD34+ progenitors were sorted with the CD34 Microbead Kit (CD34 Microbead Kit, 130-046-703, Miltenyi Biotec, Paris, France), and the 1–2.105 cells were maintained in StemMACS medium (130-100-473, Miltenyi Biotec, Paris, France) containing rh IL-3 (10 ng/mL—Miltenyi Biotec, Paris, France), rh IL-6 (10 ng/mL—Miltenyi Biotec, Paris, France), rh SCF (25 ng/mL—Miltenyi Biotec, Paris, France), rh TPO (10 ng/mL—Miltenyi Biotec, Paris, France), rh Flt-3 (10 ng/mL—Miltenyi Biotec, Paris, France), and ITS (1/100—Invitrogen, Thermo-Fisher, Illkirch, France) for 4 days in a humidified incubator at 37 °C with 5% CO_2_.


**Antibodies and reagents**


The antibodies used were anti-ABL (mouse monoclonal, 554148, BD Pharmingen, Le Pont de Claix, France), anti-actin (rabbit polyclonal, A2066, Sigma-Aldrich, Saint-Louis, MI USA), anti-GAPDH (mouse monoclonal, sc-32233, Santa-Cruz Biotec, Dallas, TX, USA), anti-phosphotyrosine (mouse monoclonal, P4110, Sigma-Aldrich, Saint-Louis, MI USA), anti-β-Tubuline (rabbit polyclonal, AB15708, Millipore, Fontenay-sous-Bois, France), anti-P-MYTP1 (sc377531, Santa-Cruz Biotec, Dallas TX, USA), anti-Rac1 (mouse monoclonal, 05-389, Millipore Fontenay-sous-Bois, France), anti-ROCK1 (mouse monoclonal, sc-17794, Santa-Cruz Biotec, Dallas TX, USA), anti-RhoA (mouse monoclonal, sc-418, Santa-Cruz Biotec, Dallas TX, USA), anti-Vav (mouse monoclonal, sc-8039, Santa-Cruz Biotec, Dallas TX, USA), and anti-P-Vav (mouse monoclonal, sc-135785, Santa-Cruz Biotec, Dallas TX, USA). The secondary antibodies used were HRP-conjugated goat anti-mouse IgG and HRP-conjugated goat anti-rabbit IgG (14231432, Fisher scientific, Thermo-Fisher, Illkirch, France).

1{β-[3-(4-methoxyphenyl)propoxyl]-4-methoxyphenethyl}-1H-imidazole hydrochloride (SKF-96365) was purchased from Sigma-Aldrich, Saint-Louis, MI, USA. Imatinib mesylate was purchased from Selleckchem (Euromedex, Souffelweyersheim, France). Mouse-EGF was acquired from Gibco, Thermo-Fisher, Illkirch, France. YM-58483, Tranilast, and GSK219 were purchased from Merck-Sigma, Saint-Louis, MI, USA; GSK-7975A was purchased from AOBIUS, Gloucester, MA, USA; and mSDF-1 was obtained from Miltenyi Biotec (Paris, France).


**Cell lysis, affinity-binding assay, immunoprecipitations, SDS–PAGE, and immunoblots**


Lysis, affinity-binding assay, immunoprecipitations SDS-PAGE, and immunoblots were performed as described previously [18]. Lysis of CD34+ progenitor cells was performed similarly to the Ba/F3 cell lines. For affinity-binding assay, sepharose-bound GST fused with RhoA-G17K (RhoA-binding GTP domain mutation) was used to determine the interactions with BCR-ABL, and sepharose-bound GST fused with Rhotekin-Rho-binding domain (RBD), or with PAK-cdc42/Rac1 interacting and binding domain (PAK-CRIB) were used to determine the presence of activated RhoA or Rac1, respectively.


**Time-lapse recordings in Matrigel**


First, 0.5 × 10^5^ cells were included in 2.5 mg/mL liquid Matrigel (356231, Corning, Corning, NY, USA) diluted in RPMI 16 40 supplemented 10% FBS at 4 °C. The mixture was then incubated for 1 h at 37 °C in a CO_2_ incubator. For the drug experiments, the cells were incubated 1 h before recording. Brightfield imaging was performed using Juli™Stage (NanoEnTek Inc., Seoul, Republic of Korea) equipment (×10 lens), and images were taken at 1 min intervals for 3 h. The tracking analysis was performed with TrackMate plugins [19].


**Leukemic Progenitor and Stem cell experiments**


LPSCs and the HS27A cell line were stained in PBS for 10 min at room temperature in the dark with 5 µM CFSE (Carboxyfluorescein succinimidyl ester—13-0850, Tonbo Biosciences, Ozyme, St Cyr-l’école, France) and 5 µM CellTraceViolet labeling (CTV—C34557, ThermoFisher, Illkirch, France), respectively. After 2 washes with PBS-5% FBS, CFSE-LPSCs were added to the adhered CTV-HS27a cell line in LPSC medium containing cytokines and 5 µM imatinib. After 3 days of co-culture, CFSE-LPSCs and CTV-HS27a were washed twice with PBS and stained with viability dye ZombieNIR (423106, Biolegend Europe BV, Amsterdam, The Netherlands) for 30 min in the dark at room temperature. After 2 washes in PBS, the cells were sorted on a BD FACS Aria III (BD Bioscience, Le Pont de Claix, France). CFSE-LPSCs were sorted according to CFSE intensity: low CFSE (high proliferation) and high CFSE (low proliferation). Each population was cultured on CTV-HS27A in LPSC medium containing cytokines with/without 5 µM imatinib and/or 40 µM SKF-96365 for 2 days. Fluorescent imaging was performed using Juli™Stage equipment (×10 lens), and images were taken at 1 min intervals for 6 h. The viability and stemness characteristics were tested for CML CD34+ cells under the different conditions for 10 days (Appendix A).


**Statistics**


The results are expressed as mean ± s.e.m. of n observations. Sets of data were compared using a Student’s *t*-test. Differences were considered statistically significant when *p* < 0.05 (ns, not significant difference; * *p* < 0.05; ** *p* < 0.01; *** *p* < 0.001). All statistical tests and graphics were performed using GraphPad Prism version 6.0 for Windows (GraphPad Software, Boston, MA, USA).

## 3. Results

### 3.1. CD34+ Cells from CML Patients Hide under Stromal HS27A Cells under Imatinib Treatment

We aimed to observe the motility behavior of CD34+ cells from CML patients in 3D co-culture with stromal HS27A cells. Recent studies showed that imatinib treatment induces quiescence of a part of leukemia progenitors and stem cells. This prompted us to focus on imatinib-induced quiescent CD34+ cells from patients. In order to specifically study low-proliferating cells, we loaded CD34+ cells with CFSE. The cells were then seeded on HS27A stromal cells for two days in control or imatinib conditions, and then sorted using FACS according to CFSE content (Figure 1a). Low-proliferating CD34+ cells were then reseeded on HS27A cells, and an analysis of the different types of motilities was performed according to [13]. We observed that 59.39% of these cells were motile in control conditions, with the large majority of them using amoeboid motility. Treatment with 5 µM imatinib (Appendix A) did not induce any change in overall motility (56.9%) nor in motility type (Appendix A). Nevertheless, we observed that low-proliferating CD34+ cells from CML patients had different types of interactions with stromal HS27A cells. During the recording time, CD34+ cells frequently moved under HS27A cells, and stayed hidden there for various lengths of time. In control conditions, 8% of CD34+ cells moved under HS27A cells, staying there for an average time of 60 min (Figure 1b,c). Treatment with imatinib induced a spectacular change in this behavior. More than 22% of cells moved under HS27A cells and stayed hidden for much longer (more than 80 min). We previously studied the motility of Ba/F3p210 cells in 3D Matrigel [13,16]. It was of interest to compare the behavior of these cells to that of CD34+ cells from CML patients in imatinib conditions. As previously described [13], these cells are spontaneously motile. They display two principal motility modes: a rolling/crawling mode characterized by an absence of cell body deformation and the use of leading-edge pseudopodia to displace the inert cell body, and an amoeboid mode in which a large pseudopodial protrusion at the leading edge directs the movement while the cell body is continuously deformed and mobilized by contraction. In contrast, Ba/F3p210S509A cells, expressing a DH/PH mutant of BCR-ABL unable to activate RhoA [17], present only a rolling/crawling mode of motility. Treatment of cells with 5 µM imatinib induced a total loss of spontaneous motility in 3D Matrigel in both cases (Figure 1d).

### 3.2. RhoA Activation Persists in Imatinib Conditions in Ba/F3p210 and CML Patients’ CD34+ Cells

Our previous reports demonstrated that the triggering of motility is dependent on Rac1, activated by Vav1, which is phosphorylated by BCR-ABL [17,20]. RhoA activation, in addition to Rac1, is obtained through the DH/PH domain of BCR-ABL [17,18] and drives the amoeboid motility mode. We first tested the Rac1 activation pathway in imatinib-treated Ba/F3p210 cells. Vav1 phosphorylation on tyrosine 174 was totally lost under imatinib treatment (Figure 2a). We then performed anti-ABL immunoprecipitation in order to highlight BCR-ABL partners. Vav1 is still present in a complex with BCR-ABL in imatinib conditions, but not under its phosphorylated form, indicating that it is not activated. We performed a Rac1 activation assay by pulldown using PAK-CRBD. As previously described, GTP-Rac1 is present in control Ba/F3p210 cells. It was not present in the imatinib-treated Ba/F3p210 cells. We were then interested in the RhoA activation pathway under imatinib conditions. We performed RTK-RBD pulldown to measure the RhoA activation state in our conditions. Whether or not imatinib is present, RhoA is still in its activated form in Ba/F3p210 cells (Figure 2b). As in Ba/F3p210 cells, RhoA activation is under the exclusive control of the DH/PH domain of BCR-ABL. We tested if this domain was still in charge of RhoA activation under imatinib treatment. We used a RhoAG17A mutant, which is unable to bind nucleotides and then prey for activated GEFs [21], to pulldown the GEF activating RhoA under imatinib conditions. As expected, the RhoAG17A mutant bound BCR-ABL under both the control and imatinib conditions (Figure 2b). Interestingly, the BCR-ABL bound to RhoAG17A was not phosphorylated under imatinib conditions, indicating that the RhoA GEF activity of BCR-ABL is tyrosine-kinase independent. We reproduced this experiment on Ba/F3p210S509A cells where BCR-ABL was not able to bind the nucleotide-free mutant of RhoA and no activation of RhoA was observed under imatinib conditions (Appendix A). We previously described that the signalplex around BCR-ABL comprises RhoA and ROCK1 in a large macromolecular complex, leading to the phosphorylation of the MyPT1 subunit of Myosin Light Chain Phosphatase [13]. We explored this signalplex under imatinib conditions using anti-ABL immunoprecipitation. ROCK1 remained in a complex with BCR-ABL, but RhoA was released from this complex. To understand if this situation modified the activity of ROCK1, we characterized via WB the phosphorylation state of MyPT1, which is a substrate of ROCK1. MyPT1 was no longer in a phosphorylated state, indicating that ROCK1, in complex with BCR-ABL, was not activated. We hypothesized that the release of RhoA from the complex under imatinib conditions separates RhoA from its effector ROCK1, preventing its activation, although RhoA is in an activated form (Figure 2c).

We then explored if the activation of RhoGTPases was comparable in CML patient progenitor cells. Interestingly, CD34+ cells from healthy donors showed no activation of RhoA, whereas CML patient CD34+ cells presented activated RhoA (Figure 2d) under the same culture conditions. These CML patient CD34+ cells still presented RhoA activation when treated with imatinib. In contrast, Rac1 was activated in cultures of healthy donor CD34+ cells and in CML patient CD34+ cells. No activation of Rac1 was observable after imatinib treatment. The pattern of RhoGTPase activation in CML patient CD34+ cells in the absence or presence of imatinib is then strictly comparable to Ba/F3p210 cells. Although RhoA is still activated under imatinib treatment, we observed that MyPT1 phosphorylation decreased according to the imatinib concentration range (Figure 2d). Nevertheless, we were unable to observe a total inhibition of MyPT1 phosphorylation, suggesting that some CD34+ cells could still present amoeboid motility. In favor of this idea, videomicroscopy analysis of the behavior of CML patient CD34+ cells in the presence or absence of imatinib showed no significant differences in motility (Appendix A).

Taken together, these results show that Rac1 is no longer activated under imatinib treatment. However, we found that RhoA remains activated, but is not linked to the BCR-ABL signalplex. This results in a loss of ROCK activation in the BCR-ABL complex and of BCR-ABL-dependent cell migration.

### 3.3. SKF-96365 Restores ROCK Signaling, Lost under Imatinib Treatment

In a previous publication, we studied the store-operated channel activity in BCR-ABL-expressing cells [16]. We were particularly interested by the use of some SOC inhibitors such as SKF-96365, since we were surprised to observe that treatment with this calcium channel antagonist was able to enhance the motility of 32dp210 cells. We then explored the effects of SKF-96365 and other SOC inhibitors such as YM-58483 and GSK-7975A, and also pharmacologic modulators of TRPV2 such as Tranilast and GSK219, in Ba/F3p210 cells. None of the inhibitors tested had any effect on Ba/F3p210 or Ba/F3p210S cell motility under control conditions or during imatinib treatment (Figure 3a and Appendix A). Under imatinib conditions, SKF-96365 did not modify the motility of these cells either, as enlightened through measurements of velocity, Euclidian distance, and directionality. We then were interested in motionless cells. Interestingly, under the imatinib and SKF-96365 condition, 27% of motionless Ba/F3p210 cells showed contractions of the cell body, without triggering motility, a phenotype that was not observed with other inhibitors (Appendix A). This phenotype was also not observed when treating Ba/F3p210S cells with imatinib and SKF-96365, suggesting that these contractions are linked, as expected, to RhoA activity. To understand the mechanism behind this phenotype, we performed BCR-ABL and RhoGTPase signaling analyses. SKF-96365 does not modify the inhibition of the tyrosine kinase activity of BCR-ABL, as shown in Figure 3b. In the same perspective, the phosphorylation state of Vav1 was also not modified by SKF-96365, nor Rac1 activation (Figure 3c). RhoA was still activated by the BCR-ABL DH/PH domain under imatinib and SKF-96365 conditions, but several changes in RhoA signaling could be observed. We found that MyPT1 was phosphorylated under imatinib and SKF-96365 conditions, indicating ROCK activation (Figure 3d). We performed anti-ABL immunoprecipitation in order to determine the presence of RhoA in complex with BCR-ABL. Surprisingly, while RhoA was released from the BCR-ABL complex under imatinib conditions, the addition of SKF-96365 allowed it to reintegrate into the complex, or prevented it from being released (Figure 3e).

### 3.4. EGF in Conjunction with SKF-96365 Restores Ba/F3p210 Cell Motility Inhibited by Imatinib

The absence of motility under imatinib conditions should be reversible, as the loss of overstimulation of multiple pathways by BCR-ABL may allow the activity of several chemoattractant receptors to be recovered [22,23]. We then tested the ability of different chemoattractants such as SDF-1 and EGF to restore the motility of Ba/F3p210 cells under imatinib treatment. None of the chemoattractants tested could restore efficient motility to Ba/F3p210 cells under imatinib conditions (Figure 4a and Appendix A). In the presence of imatinib and SKF-96365, only EGF was able to trigger the motility of Ba/F3p210 cells (Figure 4a and Appendix A).

Under imatinib conditions, EGF stimulation had no effect on the motility of Ba/F3p210 cells (Figure 4a). In contrast, EGF stimulation triggered motility of Ba/F3p210 cells in the presence of imatinib and SKF-96365, as assessed through a spider histogram showing that the cells recovered an efficient displacement comparable to the control cells. Velocity and accumulated distance were not significantly different in the imatinib, SKF-96365 and EGF conditions compared to the control (Figure 4b), whether SKF-96365 treatment alone stimulated both parameters. Interestingly, Ba/F3p210S cells were unable to recover any motility under the same conditions (Figure 4a), suggesting that RhoA activation by BCR-ABL is necessary for SKF-96365’s action. The hypothesis of EGF triggering the tyrosine phosphorylation of BCR-ABL was rejected via a Western blot showing no difference in BCR-ABL phosphorylation between the imatinib and imatinib, SKF-96365 and EGF conditions (Figure 4c). Concerning the activation of RhoGTPases, no difference was observed for RhoA when adding EGF to imatinib and SKF-96365 (Figure 4d), and ROCK signaling was conserved as assessed through MyPT1 phosphorylation. Conversely, we observed the presence of GTP-Rac1 under these conditions (Figure 4e). It must also be noted that an efficient motility of Ba/F3p210 cells was restored (Figure 4a). Surprisingly, Vav1 was not phosphorylated, indicating that another GEF is responsible for Rac1 activation. Interestingly, EGF was unable to reactivate Rac1 in Ba/F3p210S509A cells under the imatinib + SKF-96365 conditions (Appendix A), indicating the necessity of RhoA activation by BCR-ABL prior to observing the effects of SKF-96365 (Appendix A).

### 3.5. SKF-96365 Expels Low-Proliferating CD34+ Cells from the HS27A Niche

The CFSE-rich population (see Figure 1a) sorted after two days of culture in the presence or absence of imatinib was then seeded on human stromal HS27A cells (Figure 5a). Each sample was cultured for two more days in the presence of imatinib with or without SKF-96365. We analyzed the motility of these sorted CFSE-rich CD34+ cells in co-culture with HS27A cells. No clear difference was present in the 2D motility of these cells in the presence or absence of SKF-96365 with imatinib (Figure 1b). However, concerning CD34+ cells slipping under HS27A cells, we observed a significant modification when SKF-96365 was present.

We determined the different behaviors of CD34+ cells moving below HS27A cells, (i.e., cells having back and forth movements and staying below HS27A cells for less than 30 min; cells that stayed below HS27A cells for more than 30 min to more than 300 min). We also measured the average time spent below the HS27A cells. In CFSE+ CD34+ cells, 6.17% of the analyzed cells stayed hidden under HS27A cells for an average of 104 min under control conditions (Figure 5b). Under imatinib conditions, this proportion raised to 11.11% of total cells for an average of 68 min. The addition of SKF-96365 completely changed these behaviors. A proportion of 1.54% of cells moved under HS27A cells and stayed hidden for an average of 21 min under control SKF-96365 conditions. Under imatinib and SKF-96365 conditions, only 0.8% of the cells moved under HS27A cells for an average of 13 min. Interestingly, the analysis of the behavior of these hiding cells also showed a difference between the control and the imatinib- and SKF-96365-treated cells. We sorted the behavior of hiding cells into those with back-and-forth displacements under HS27A cells—short duration: SD—and those that stayed under HS27A cells for longer than 30 min—long duration: LD (see Appendix A for details). Under the control and imatinib conditions, 86.6% and 77.6% of hiding CFSE+ CD34+ cells displayed LD behavior, respectively. We only observed SD behavior in the SKF-96365-treated cells (Figure 5c). SKF-96365 clearly inhibited the ability of low-proliferating CD34+ cells from CML patients to move and stay below stromal HS27A cells.

## 4. Discussion

The treatment of CML with TKIs has allowed hundreds of thousands of patients to attain long-term molecular remission, although the eradication of tumoral clones remains difficult. TKIs are very efficient in differentiating CML progenitors and differentiated hematopoietic cells but can paradoxically trigger quiescence of CML hematopoietic stem and unengaged progenitor cells [4,5]. These quiescent LSCs remain in the hematopoietic niche for years and allow resurgence of the disease if imatinib treatment is interrupted [2]. Interestingly, TKIs only trigger quiescence in BCR-ABL-expressing cells, suggesting that this effect relies on a tyrosine-kinase-independent BCR-ABL activity. Given the importance of quiescent cells during imatinib treatment, we used CFSE to sort low-proliferating CML CD34+ cells under imatinib treatment. In a reconstructed stem niche, we observed that imatinib-treated cells very frequently slip under HS27A cells and stay hidden below them for long periods, a behavior that is less observed in untreated cells. It must be noted that unsorted CML CD34+ cells showed no obvious difference in motility when imatinib was used.

Imatinib induces a total loss of motility in Ba/F3p210 cells. We observed the same effect on 32dp210 cells. In recent years, we have studied the activation of RhoGTPases by BCR-ABL and the motility of CML cells [13,17,18]. It was then of interest to determine the status of RhoGTPases in the presence of TKI treatment. Loss of motility in Ba/F3p210 cells suggested that RhoGTPases are in an inactive form. Imatinib indeed induced total inhibition of Rac1 activation, which is highly activated in untreated Ba/F3p210 cells. We previously showed that Rac1 activation was under the dependence of Vav1 GEF in these cells, itself being activated by tyrosine phosphorylation via BCR-ABL [18,20]. As expected, Vav1, although still bound to BCR-ABL, was no more tyrosine-phosphorylated under imatinib conditions. Rac1 activation in BCR-ABL-expressing cells is thus dependent on the tyrosine kinase activity of BCR-ABL.

In contrast, RhoA was still in its GTP-bound form in the imatinib-treated Ba/F3p210 cells. Using the activated GEF-specific RhoAG17V mutant, we show that this GEF activity is attributable to the DH/PH domain of BCR-ABL. In addition, in GEF-mutated Ba/F3p210S509A cells, no extrinsic RhoA-GEF activity was observed under imatinib conditions. The RhoA-GEF activity of BCR-ABL is thus independent of its tyrosine kinase activity. Paradoxically, although RhoA was still activated under imatinib treatment, ROCK1, still bound to BCR-ABL, was unable to phosphorylate MyPT1 under these conditions. This suggests that the RhoA/ROCK pathway is not activated under imatinib treatment, whether RhoA is in its GTP-bound form.

In a previous report, we focused on store-operated calcium channel regulation by BCR-ABL. We reported an inhibition of store-operated calcium entry and calcium transients through the expression of p210BCR-ABL in 32d cells resulting in a decrease in NFAT activation [16]. Surprisingly we also observed an enhancement of motility in 32dp210 cells using the SKF-96365 inhibitor. None of the other SOC or calcium channel inhibitors tested were able to enhance the motility of Ba/F3p210 cells in either the control or imatinib conditions. Interestingly, under imatinib conditions, SKF-96365 induces contractions of the cell body without triggering motility, a phenotype that was not observed using other calcium channel inhibitors. This suggests that the effect of SKF-96365 is not dependent on its effect on SOC or on other channels such as TRPC or TRPV2. These contractions are BCR-ABL and RhoA dependent, since they are not observed in Ba/F3p210S509A cells. In fact, the ROCK/MLC pathway was restored under imatinib + SKF-96365 conditions, corresponding to the apparition of contractions without migration. This surprisingly suggests that SKF-96365 allows RhoA to transduce signals to ROCK1 in the presence of imatinib. Moreover, it has to be noted that the contractions induced by SKF-96365 last over 8 h in imatinib-treated Ba/F3p210 cells.

Several studies suggest that part of the effects of SKF-96365 could be related to an effect on Akt [24,25]. However, in Ba/F3p210 cells, no difference in Akt phosphorylation could be observed between the control, imatinib, SKF-96365, or imatinib and SKF-96365 treatments (Appendix A). It seems unlikely that Akt is involved in the activation of the ROCK1 pathway here. SKF-96365 is a nonspecific inhibitor of SOCE, also known as a TRPC blocker, and was shown to inhibit transient receptor potential vanilloid 2 (TRPV2) channels at high concentration (250 μM) [26]. It was also shown that 20 µM of SKF-96365 inhibits TRPM7 [27], which is a bifunctional protein comprising a TRP ion channel segment linked to an α-type protein kinase domain [28]. It is possible that the effect of SKF-96365 on BCR-ABL-expressing cells could be due to an effect on channel proteins, independently of channel function and calcium entry, but could be related to the scaffolding ability of these proteins. For instance, TRPCs have a direct interaction with CaM, and biochemical evidence has also been provided for an association with G proteins and scaffolding proteins such as Homer1, RACK1, EB-50, Ezrin, NHERF, and caveolin [29]; an interaction of TRPC1 with RhoA has also been shown [30]. The effects of SKF-96365 were also tested in other malignant diseases, unveiling different types of action depending on the tumor. In glioblastoma cells, it inhibited the growth of cultured human GB cells [31]. This effect was clearly dependent on the inhibition of both TRPC channel activity and NCX. In GMB stem cells isolated from patients, the compound also inhibited proliferation and self-renewal [32]. In colorectal cancer, SKF96365 induced cell cycle arrest and apoptosis [25]. Here, the effect of SKF96365 was obtained by inhibiting the CAMKII/AKT signaling cascade. Finally, in bladder cancer, the combination of oxiplatin and SKF96365 enhanced the apoptosis of cancer cells [33]. Again, the effect was Ca^2+^ channel blocking dependent. Further research must be developed to explore these hypotheses.

In imatinib and SKF-96365-treated Ba/F3p210 cells, EGF restores motility in a manner comparable to the control conditions (Figure 4a). This effect is linked to the activation of Rac1, which we previously demonstrated to be the motility-triggering signal in these cells [17]. Under these conditions, the activation of Rac1 is independent of Vav1, which remains unphosphorylated, although still in complex with BCR-ABL. Another GEF should promote this activation of Rac1. Interestingly, EGF could not induce motility in imatinib-treated cells. Also, in imatinib and SKF-96365-treated Ba/F3p210S509A cells, which do not activate RhoA, EGF was unable to trigger efficient motility, and no Rac1 activation was observed. This suggests that EGF signaling to Rac1 necessitates the presence of active RhoA. It represents an important change in paradigm, because in Ba/F3p210 cells, we demonstrated an uncoupling of Rac1 signaling triggering motility and RhoA signaling inducing amoeboid-type motility. In contrast, we suggest that SKF-96365 redistributes the signal around BCR-ABL in the presence of imatinib, generating new RhoGTPase pathways to trigger motility in which Rac1 activation is dependent on RhoA activation.

The effects of SKF-96365 restoring the motility of Ba/F3p210 cells under imatinib treatment led us to explore its effects on imatinib-treated progenitors from CML patients. Using CD34+ cells purified from patient samples taken at diagnosis, we could observe that the activation of RhoGTPases was totally comparable to Ba/F3p210 cells. Rac1 and RhoA were in a GTP-bound state under basal conditions, and imatinib treatment inhibited Rac1 activation while RhoA was still in its activated form. A significant inhibition of MyPT1 phosphorylation was also observed under these conditions, suggesting inhibition of the ROCK pathway.

Interestingly, while the mobility of CD34+ cells from patients was not globally affected by imatinib treatment, an important proportion of low-proliferating CD34+ cells tended to slip and hide under HS27A cells. According to the literature describing the emergence of quiescence for CML stem/progenitor cells under imatinib treatment, this suggests that these hiding cells could represent a quiescent population. In accordance with the results obtained on Ba/F3p210 cells, SKF-96365 induces a reversion of this hiding phenotype, and restores full motility to these cells, which then leave the protective layer provided by HS27A.

It has to be noted that inhibition of ROCK by Y-27632 negatively affects the expansion of CD34+ hematopoietic progenitor cells [14]. Here, SKF-96365 restores ROCK signaling and helps CML stem and progenitor cells to escape from the niche. The importance of the RhoA/ROCK pathway in the context of the imatinib resistance of CML stem cells is to be reconsidered.

## 5. Conclusions

Altogether, these results suggest that SKF-96365 or derived molecules could be interesting compounds to target quiescent CML stem and progenitor cells under TKI treatment. Taking into account that these CML LSCs are protected in the leukemic niche, forcing them to migrate out of the niche with SKF-96365 may push them into the cell cycle and render them sensitive to TKIs again.

## Figures and Tables

**Figure 1 cancers-16-02791-f001:**
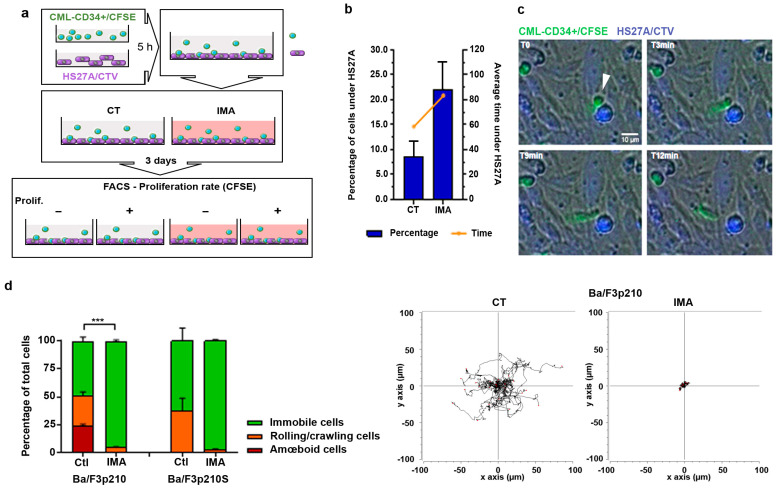
**Imatinib treatment modifies CML-patient CD34+ and Ba/F3p210 cell mobility.** (**a**) CML patients’ CD34+ cells and HS27A cells were stained with 5 µM CFSE and 5 µM CTV, respectively. After 5 h, CFSE-stained CD34+ cells were added to adherent CTV-HS27A cells with or without 5 µM imatinib. After 3 days of co-culture, CFSE-CD34+ and CTV-HS27A were sorted according to CFSE intensity: low CFSE (high proliferation) and high CFSE (low proliferation). (**b**) The percentage of CD34+ cells hiding under HS27A and the average time spent under HS27A were determined through a time-lapse videomicroscopy analysis (N = 2 patients, n = 5). (**c**) Time-lapse images (t = 0; t = 3 min; t = 9 min and t = 12 min) extracted from JuliStage recordings of CFSE-stained CD34+ cells from CML patients seeded on CTV-stained HS27A stromal cells in imatinib (5 µM) conditions. Scale bar: 10 µm. Arrowhead shows CD34+ cell hiding under HS27A cells. (**d**) Cells (0.5 × 10^5^ Ba/F3p210 and Ba/F3p210S509A) were included in 2.5 mg/mL liquid Matrigel diluted in culture medium and cells were incubated with 5 µM imatinib. Each type of displacement (immobile cells, rolling/crawling, or amœboid displacement) was determined through time-lapse videomicroscopy analysis (n > 3). Right panel: spider graph showing the accumulated tracks of analyzed individual cell motility for Ba/F3p210 cells treated or not with imatinib 5 µM. *t*-test *** *p* ≤ 0.001.

**Figure 2 cancers-16-02791-f002:**
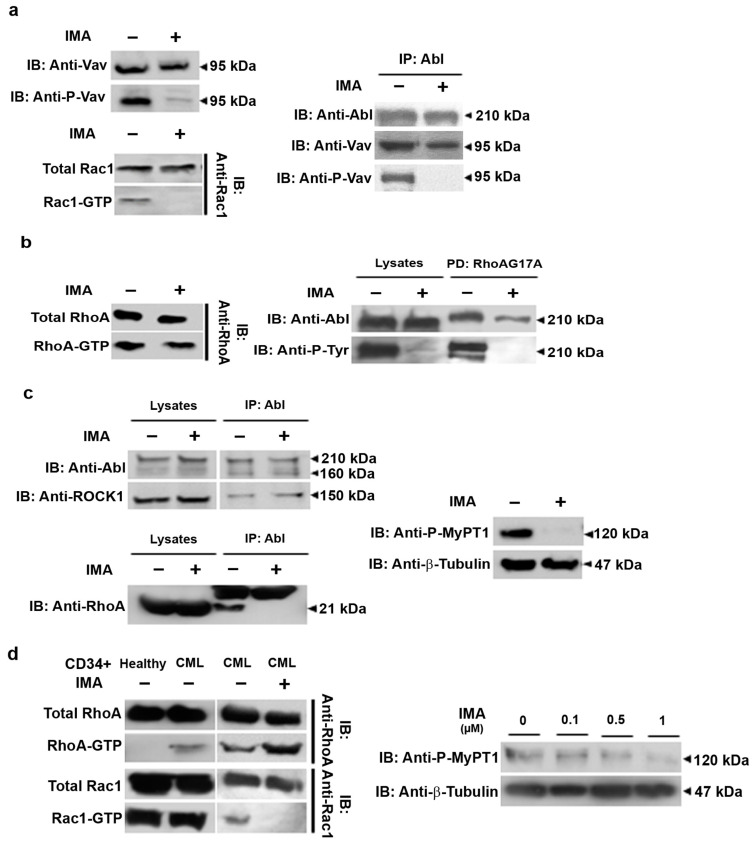
**RhoGTPase signaling in imatinib conditions**. (**a**) Western blot analysis of phosphorylated and total Vav, activated (GTP-Rac1), and total Rac1 in Ba/F3p210 cells. Coimmunoprecipitation of Vav and P-Vav, with p210^BCR-ABL^. Ba/F3p210 cell lysates were immunoprecipitated using anti-ABL IgG and immunoblotted as indicated. Western blot analysis of activated (GTP-Rac1) and total Rac1 in Ba/F3p210 cells obtained via GST-pulldown. (**b**) Western blot analysis of activated (GTP-RhoA) and total RhoA in Ba/F3p210 cells obtained via GST-pulldown. Affinity-binding assay of phosphorylated and total p210^BCR-ABL^ with negative dominant RhoA (RhoAG17A). Ba/F3p210 cell lysates were immunoprecipitated with sepharose-bound GST-fused with RhoA-G17A and immunoblotted as indicated. (**c**) Coimmunoprecipitation of ROCK1 and RhoA with p210^BCR-ABL^. Ba/F3p210 cell lysates were immunoprecipitated with anti-ABL IgG and immunoblotted as indicated. Western blot analysis of phosphorylated MYPT1 and tubulin in Ba/F3p210 cells. (**d**) Western blot analysis of activated (GTP-RhoA or GTP-Rac1) and total RhoA or Rac1 in CML patient CD34+ cells obtained via GST-pulldown. Western blot analysis of phosphorylated MYPT1 and β-tubulin in CML patient CD34+ cells. Each experiment was carried out 3 times. Ba/F3p210 and CD34+ patient cells were treated with 1 µM ((**d**) top), 5 µM imatinib (**a**–**c**), or as indicated (**d**).

**Figure 3 cancers-16-02791-f003:**
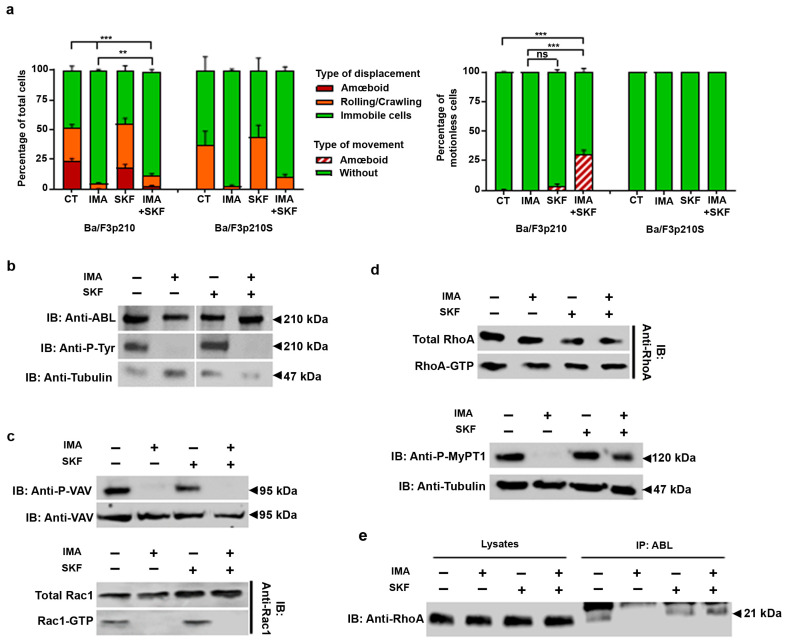
**SKF-96365 induces amœboid movements in immobile imatinib-treated Ba/F3p210 cells.** (**a**) Cells (0.5 × 105 Ba/F3p210 and Ba/F3p210S509A) were included in 2.5 mg/mL liquid Matrigel diluted in culture medium and incubated with or without 5 µM imatinib and/or 40 µM SKF-96365. Each type of displacement (immobile cells, rolling/crawling, or amœboid displacement) was determined through time-lapse videomicroscopy analysis. The right panel shows the analysis of amoeboid contractions in immobile cells (n > 3). *t*-test ** *p* ≤ 0.01; *** *p* ≤ 0.001; ns: non-significant. (**b**) Western blot analysis of phosphorylated and total p210^BCR-ABL^ in Ba/F3p210 cells during imatinib and SKF-96365 treatment. (**c**) Western blot analysis of phosphorylated and total Vav, activated (GTP-Rac1), and total Rac1 in Ba/F3p210 cells during imatinib and SKF-96365 treatment. (**d**) Upper panel: Western blot analysis of activated (GTP-RhoA) and total RhoA or Rac1 in Ba/F3p210 cells obtained via GST-pulldown. Lower panel: Western blot analysis of phosphorylated MYPT1 and β-tubulin in CML patient CD34+ cells. (**e**) Coimmunoprecipitation of RhoA with p210^BCR-ABL^. Ba/F3p210 cell lysates were immunoprecipitated with anti-ABL IgG and immunoblotted as indicated.

**Figure 4 cancers-16-02791-f004:**
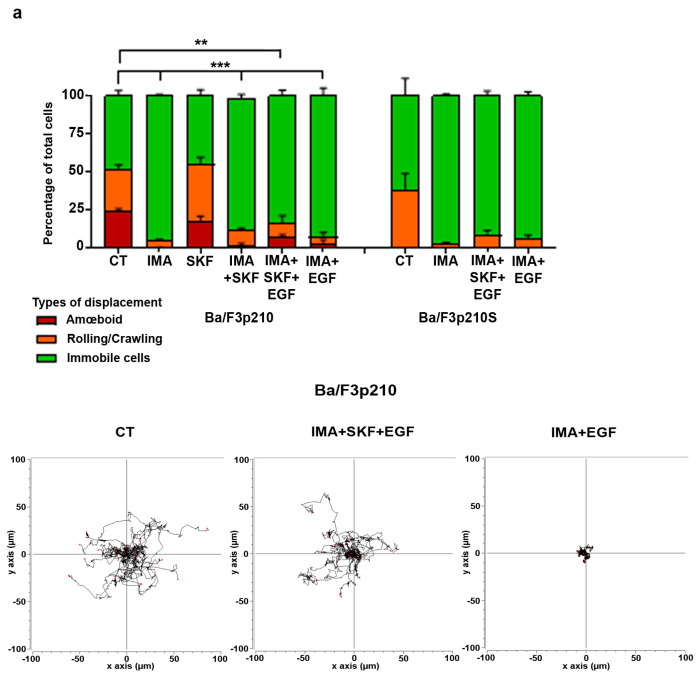
**EGF in conjunction with SKF-96365 restores Ba/F3p210 cell motility inhibited by imatinib.** (**a**) Cells (0.5 × 10^5^ Ba/F3p210 and Ba/F3p210S509A) were included in 2.5 mg/mL liquid Matrigel diluted in culture medium and incubated with or without 5 µM imatinib and/or 40 µM SKF-96365 and/or EGF 10 ng/mL. Each type of displacement (immobile cells, rolling/crawling, or amœboid displacement) was determined through time-lapse videomicroscopy analysis (n > 3). Right panel: spider graph showing the accumulated tracks of analyzed individual cell motility for Ba/F3p210 cells treated or not with imatinib 5 µM. *t*-test ** *p* ≤ 0.01; *** *p* ≤ 0.001. (**b**) Analysis of velocity (µm/s) (left) and accumulated distance (µm) (right) of Ba/F3p210 cells treated with SKF-96365 alone or imatinib+ SKF-96365 and EGF. Data were obtained using TrackMate plugins. * *p* < 0.05; ** *p* < 0.01; ns: non-significant. (**c**) Western blot analysis of phosphorylated p210^BCR-ABL^ in Ba/F3p210 cells under the indicated conditions. (**d**) Upper panel: Western blot analysis of activated (GTP-RhoA) and total RhoA in Ba/F3p210 cells obtained via GST-pulldown. Lower panel: Western blot analysis of phosphorylated MYPT1 and actin in Ba/F3p210 cells. (**e**) Western blot analysis of activated (GTP-Rac1) and total Rac1, and phosphorylated and total Vav in Ba/F3p210 cells under the indicated conditions.

**Figure 5 cancers-16-02791-f005:**
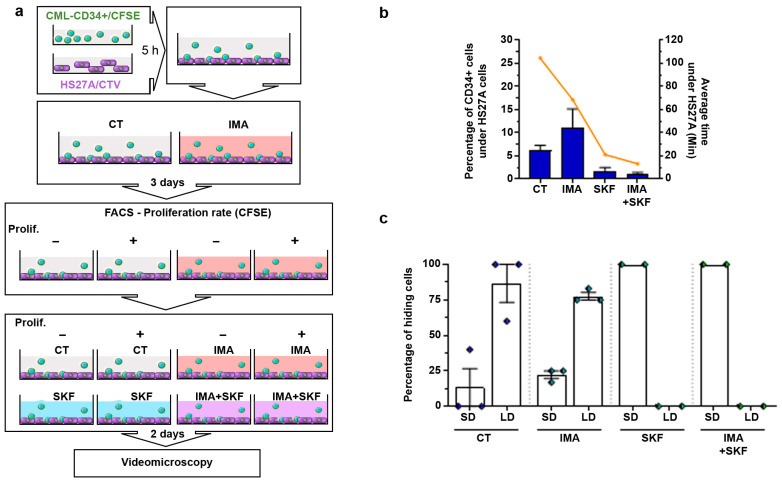
**SKF-96365 expels low-proliferating CD34+ cells from the HS27A niche.** (**a**) CML patient CD34+ cells and HS27A cells were stained with 5 µM CFSE and 5 µM CTV, respectively. After 5 h, CFSE-stained CD34+ cells were added to adherent CTV-HS27A cells with or without 5 µM imatinib. After 3 days of co-culture, CFSE-CD34+ and CTV-HS27A were sorted according to CFSE intensity: low CFSE (high proliferation) and high CFSE (low proliferation). CFSE-sorted CD34+ cells from CML patients were reseeded on previously cultured CTV-stained HS27A cells and cultured for 2 days under the indicated conditions. Then, videomicroscopy was performed using JuliStage in fluorescent mode during 6 h recording. (**b**) The percentage of CD34+ cells hiding under HS27A cells and the average time spent under HS27A cells were determined through time-lapse videomicroscopy analysis (N = 2 patients, n = 5). (**c**) Analysis of the hiding behavior of low-proliferating CD34+ cells from CML patients during the recording time. Two behaviors were identified corresponding to a short-duration stay under HS27A cells (SD; <30 min) and a long-duration stay (LD; 30 to >300 min) (see Appendix A for details).

## Data Availability

All data are contained within the manuscript.

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
