# Peer review of "SKF-96365 Expels Tyrosine Kinase Inhibitor-Treated CML Stem and Progenitor Cells from the HS27A Stromal Cell Niche in a RhoA-Dependent Mechanism"

_cancers, 2024, doi:10.3390/cancers16162791_

Round 1
Reviewer 1 Report
Comments and Suggestions for Authors
The introduction of tyrosine kinase inhibitors (TKIs) in the treatment of chronic myeloid leukemia (CML) has been highly successful. However, persistent leukemic stem cells (LSCs) continue to pose a significant challenge, preventing the complete cure of most CML patients. These persistent cells often remain in a non-proliferative, quiescent state, and evidence suggests that TKI therapy may support this quiescence.
Given the observation that quiescent stem cells exhibit reduced motility the authors observed that non-proliferating, TKI-treated LSCs "hide" beneath HS27A stromal cells in a manner dependent on RhoA-GTP activity, while Rac1 is inhibited under these conditions. The authors propose that this phenomenon results from the loss of ROCK activation due to RhoA release from the Bcr-Abl-RhoA-ROCK complex. By using a calcium channel inhibitor, SKF96365, LSCs regained motility and exited the stromal niche.
Although the concept is highly relevant and intriguing, additional experiments are required to fully substantiate the presented hypothesis.
Major Points:
The authors show that the combination of TKI and SKF96365 induces egress from the stromal niche. However, what impact does this have on the long-term survival of CD34+ cells? Is targeting LSCs via calcium blockade superior to TKI monotherapy in terms of effectiveness?
It is not clear if the behavior of CML CD34+ cells in the model used is indeed linked to TKI persistence. It would be beneficial to demonstrate whether LSCs hiding under stromal cells exhibit greater resistance to TKI therapy. For example, what are the effects on apoptosis or quiescence in LSCs treated with TKI while hiding or not hiding under HS27A stromal cells?
The rationale behind the specific use of HS27A stromal cells should be clarified. Could similar results be obtained using other stromal cell types, such as HS5?
The manuscript lacks experiments addressing the mode of action of SKF96365. How is the effect of SKF96365 on calcium channel inhibition confirmed?
Minor Points:
The authors suggest that SKF96365 could be an effective compound for targeting quiescent LSCs. Are there any existing data on the in vivo performance of such inhibitors in malignant diseases, and how do they compare in effectiveness? Please discuss this aspect.
In line 232, the authors mention that CD34+ cells from healthy donors did not show RhoA activation. However, only one sample from a healthy donor is presented. Increasing the number of samples would strengthen this observation.
The text formatting in line 329 needs to be corrected.
Comments on the Quality of English LanguageWhile the manuscript provides valuable insights, there are instances where minor improvements in the English language could significantly enhance clarity and comprehension. Refining the phrasing and grammar in certain sections would make the text more accessible and easier to understand for a broader audience.
Reviewer 2 Report
Comments and Suggestions for Authors
Review comments for “SKF-96365 expels Tyrosine Kinase Inhibitor-treated CML stem and progenitor cells from HS27A stromal cell niche in a RhoA-dependent mechanism.”
Major comments:
1. Westernblot: Some of your westernblot results don’t have tubulin band to show the total protein level are equal between groups, for example, figure 2a and b. It would be beneficial to explain it.
2. Lines 406-414: You mention the paradoxical effect of imatinib on CML stem and progenitor cells leading to quiescence. Clarify why quiescence is a challenge in the treatment of CML and how it affects long-term outcomes. Adding a sentence or two on the clinical implications of quiescent cells could provide context for readers less familiar with the subject.
3. Clinical Implications: The potential therapeutic implications of your findings are highly significant. Elaborate on how restoring motility in quiescent CML cells might impact treatment strategies. Discuss any future research directions or potential clinical applications that could arise from your findings.
Minor comments:
1. Grammer error: The sentence “Low proliferating CD34+ cells were then reseeded on HS27A cells, and analysis of the different types of motility was performed, according to (13)”. Is misleading. It would be beneficial to change it to “Low proliferating CD34+ cells were then reseeded on HS27A cells, and analysis of the different types of motilities were performed, according to reference 13.” And review the text for grammatical errors and ensure that sentences are well-constructed and easy to follow.
Comments on the Quality of English LanguageGrammer error: The sentence “Low proliferating CD34+ cells were then reseeded on HS27A cells, and analysis of the different types of motility was performed, according to (13)”. Is misleading. It would be beneficial to change it to “Low proliferating CD34+ cells were then reseeded on HS27A cells, and analysis of the different types of motilities were performed, according to reference 13.” And review the text for grammatical errors and ensure that sentences are well-constructed and easy to follow.
Author Response
- Westernblot:Some of your westernblot results don’t have tubulin band to show the total protein level are equal between groups, for example, figure 2a and b. It would be beneficial to explain it.
We presented tubulin or GAPD or actin charge controls every time it was necessary. In pulldown experiments, analyzed proteins are their own control, as we compare pair by pair (Lysate and Pulldown) the quantities of each protein. In our past publications, we determined by densitometry the amount of RhoGTPase in an activated state. Here, we did not realized these measurements, as it is black and white, there is an activated form of RhoA or Rac1, or not. We did not observed any variation of RhoA activation state in the different conditions. The same answer can be made for IPs in which we compare phosphorylated form to total form of the same protein.
- Lines 406-414: You mention the paradoxical effect of imatinib on CML stem and progenitor cells leading to quiescence. Clarify why quiescence is a challenge in the treatment of CML and how it affects long-term outcomes. Adding a sentence or two on the clinical implications of quiescent cells could provide context for readers less familiar with the subject.
Thank you for giving us the opportunity to clarify the context of the study. We modified the text in the concerned paragraph as requested.
- Clinical Implications: The potential therapeutic implications of your findings are highly significant. Elaborate on how restoring motility in quiescent CML cells might impact treatment strategies. Discuss any future research directions or potential clinical applications that could arise from your findings.
Thank you again for this interesting remark. We modified the conclusion as requested, to point out the importance of motility on quiescent cells.
Minor comments:
- Grammer error:The sentence “Low proliferating CD34+ cells were then reseeded on HS27A cells, and analysis of the different types of motility was performed, according to (13)”. Is misleading. It would be beneficial to change it to “Low proliferating CD34+ cells were then reseeded on HS27A cells, and analysis of the different types of motilities were performed, according to reference 13.” And review the text for grammatical errors and ensure that sentences are well-constructed and easy to follow.
Thank you for pointing this mistake. We modified in the text as requested. We also will have an English reviewing through the service provided by MDPI.

Reviewer 3 Report
Comments and Suggestions for Authors
In this study, Dubourg et al. tried to find alternative ways to eradicate Chronic Myeloid Leukemia (CML) Leukemia Stem Cell (LSC). The authors showed that low proliferating CD34+ cells from CML patients in 3D co-culture hide under HS27A stromal cells during Tyrosine kinase inhibitors (TKI) treatment, a behavior less present in untreated cells. In the same conditions, Ba/F3p210 cells lose their spontaneous motility. In CML CD34+ and in Ba/F3p210 cells, while Rac1 is totally inhibited by TKI, RhoA remains in activated form. Then, the authors exhibited that co-incubation of TKI with SKF-96365 induced expulsion of these cells from the HS27A niche. Together, this underscores a role for RhoA in LSC behavior under TKI treatment and suggests that SKF-96365 could remobilize quiescent CML LSCs through reactivation of RhoA/ROCK pathway.
One of the major problems encountered in CML treatment is that some LSC and immature progenitors are insensitive to TKIs and persist into the hematopoietic niche, representing a pool of malignant cells responsible for disease recurrence in case of TKI withdrawal. No study has examined the role of RhoA/ROCK pathway in CML LSCs. Therefore, this reviewer thinks that the data presented include novel findings and would be beneficial for readers in community.However, the claims are not fully supported by the experimental data and the conclusion is held by the results derived from one TKI imatinib. Therefore, the experimental evidence provided in this manuscript is limited. This reviewer has the following specific critiques that may help to improve the overall manuscript.
1. The conclusion is held by the results obtained from one TKI, imatinib. This reviewer thinks that it is insufficient. The authors should reproduce Figures 1, and 2 using at least one or two more TKIs, such as nilotinib or dasatinib.
2. Same as above, the conclusion is held by the results obtained from one cell line, BaF3p210. This reviewer thinks that it is insufficient. The authors should reproduce several data shown in Figures 2-4 using at least one or two more cell lines, such as 32Dp210.
Author Response
We agree with the reviewer that studying other TKIs and other cell lines would strenghthen this manuscript. However, his demands corresponds to a full new study that would require for us one or two years of work.
Round 2
Reviewer 1 Report
Comments and Suggestions for Authors
I have nor further comments.
Author Response
Thank you to this reviewer. No further need for an answer
Reviewer 3 Report
Comments and Suggestions for Authors
Unfortunately, the authors did not respond this reviewers' suggestions at all in the revised manuscript. This reviewer thinks that the claims are not supported by the experimental data.
Author Response
I think that the reviewer's demands are not reasonable. In fact, We do not understand why the use of 32D cells would add anything to the meaning of our results. We demonstrated that CD34+ cells from patients have a similar behavior than Ba/F3 cells and showing the same results on 32D cells would not give significant enhancement of the global message. Concerning the other TKIs suggested by the reviewer, we could make the experiments if mandatory from the editor, but it would take a lot of time, probably a full year for obtaining solid results, without any assurance that these results will add anything to the message already present.
This is why we do not wish to answer positively to this reviewer's demands.